# Health postservice readiness and use of preventive and curative services for suspected childhood pneumonia in Ethiopia: a cross-sectional study

Amare Tariku [1,2] Yemane Berhane,[2] Alemayehu Worku,[2,3]
Gashaw Andargie Biks,[4] Lars Åke Persson [5,6] Yemisrach Behailu Okwaraji[5,6]

For numbered affiliations see end of article.

**Correspondence to**
Dr. Amare Tariku;
amaretariku15@yahoo.com

## ABSTRACT

**Objective** Pneumonia is the single-leading cause of infectious disease deaths in children under-5. Despite this challenge, the utilisation of preventive and curative child health services remains low in Ethiopia. We investigated the association between health post service readiness and caregivers' awareness of pneumonia services, care-seeking and utilisation of pneumonia-relevant immunisation in four Ethiopian regions.

**Design and setting** This cross-sectional study was conducted in 52 districts of four regions of Ethiopia from December 2018 to February 2019. The health posts preparedness for sick child care was assessed using the WHO Health Service Availability and Readiness Assessment tool. Multilevel analyses were employed to examine the associations between health post readiness and household-level awareness and utilisation of services.

**Participants** We included 165 health posts, 274 health extension workers (community health workers) and 4729 caregivers with 5787 children 2–59 months.

**Outcome measures** Awareness of pneumonia treatment, care-seeking behaviour and coverage of pentavalent-3 immunisation.

**Results** Only 62.8% of health posts were ready to provide sick child care services. One-quarter of caregivers were aware of pneumonia services, and 56.8% sought an appropriate care provider for suspected pneumonia. Nearly half (49.3%) of children (12–23 months) had received pentavalent-3 immunisation. General health post readiness was not associated with caregivers' awareness of pneumonia treatment (adjusted OR, AOR 0.9, 95% CI 0.7 to 1.1) and utilisation of pentavalent-3 immunisation (AOR=1.2, 95% CI 0.8 to 1.6), but negatively associated with care-seeking for childhood illnesses (AOR=0.6, 95% CI 0.4 to 0.8).

**Conclusion** We found no association between facility readiness and awareness or utilisation of child health services. There were significant deficiencies in health post preparedness for services. Caregivers had low awareness and utilisation of pneumonia-related services. The results underline the importance of enhancing facility preparedness, providing high-quality care and intensifying demand generation efforts to prevent and treat pneumonia.

## Strengths and limitations of this study

► This is the first study, which assessed the association between first-level healthcare facility readiness for services and caregivers' awareness and utilisation of pneumonia-relevant preventive and curative services in four of the most populous Ethiopian regions.

► Facility preparedness was assessed using the WHO Health Service Availability and Readiness Assessment tool to generate objective and reliable information that is comparable across or within countries.

► We ascertained information on the utilisation of pneumonia-related preventive and curative services with Demographic and Health Surveys methods, assumed to reduce recall bias.

► The study covered pneumonia services and service readiness of health posts in 52 districts of four Ethiopian regions and findings may not be representative of other geographic areas and health system levels.

## INTRODUCTION

Worldwide, pneumonia is responsible for one-fifth of all under-5 deaths.[1] Nearly half (49%) of these deaths occur in four low-income and middle-income countries, including Ethiopia.[2 3] There are effective preventive measures, including immunisation and antibiotics that can prevent or treat most cases of pneumonia. Despite these resources, children in low-income countries continue to suffer and die from pneumonia due to lack of access to or availability of these services.[4]

The WHO and UNICEF introduced the integrated Community Case Management (iCCM) strategy in 2004 to increase access and quality of services for childhood pneumonia, diarrhoea and malaria. Effective implementation of this strategy requires uninterrupted stock of drugs and supplies, trained community health workers, and

community awareness of these services.[5 6] Nevertheless, studies in low-income countries reveal that less than one-fifth of sick children were brought to relevant health services for suspected pneumonia.[7 8] This low level of care-seeking has, among other things, been attributed to the poor quality of health services.[9] A majority of facilities in these settings have reportedly not had essential drugs, supplies and trained community health workers.[10–13] The general service readiness index for sick child care has varied between 19% and 69% in reports from sub-Saharan African countries.[11 13] The readiness has been lower in rural areas and at the lowest primary healthcare level.[14–17] In Ethiopia, earlier reports have shown a lack of iCCM drugs and supplies at health posts.[9 18]

Inadequate preparedness and low service quality at the primary healthcare level reduce parents' trust and utilisation of health services.[11 19–21] In Ethiopia, the low utilisation of iCCM services was also attributed to absent supervision and refresher training of health extension workers.[22–27] We have earlier shown that a substantial proportion of caregivers were not aware of pneumonia-related health services and, therefore, less likely to seek care when their children got sick or get their children immunised.[28 29] The low utilisation of iCCM services has also been attributed to the lack of readiness of health posts to care for sick children.[19 30 31] Thus, there are reasons to investigate the primary-level health facility preparedness to provide child health services and whether this is associated with the coverage of pneumonia-related preventive and treatment services. In Ethiopia, under-5 mortality was reported to be 55 per 1000 live births in the 2019 Demographic and Health Survey. Although reduced in recent decades, the persistently high level and continued pneumonia deaths call for intensified efforts to prevent these preventable deaths.[28 32] The Ethiopian Ministry of Health in collaboration with non-governmental organisations implemented a complex community-based intervention labelled Optimising the Health Extension Programme (OHEP) in four of the most populous regions. The OHEP aimed to contribute to reductions in under-5 child mortality through enhancing utilisation of iCCM and community-based newborn care services.[33] This study was part of the evaluation of OHEP intervention. We aimed to examine the association between the health post readiness to serve and caregivers' awareness of pneumonia-related services, care-seeking for sick children, and whether their 12–23 months old children had got three pentavalent vaccinations.

## METHODS
### Study setting and design
The Ethiopian primary healthcare system typically consists of a primary hospital, a health centre and five satellite health posts. A health post is the lowest service delivery point staffed by two health extension workers serving around 5000 rural residents. Since 2003, Ethiopia has implemented the health extension programme

to achieve universal coverage of primary healthcare for the rural population. This national programme is implemented by health extension workers, and they provide basic promotive, preventive and curative services through outreach and health post-based approaches. In 2010, after a change in policy that allowed the health extension workers to treat child pneumonia, the Ethiopian Ministry of Health and partners initiated the implementation of iCCM of childhood illnesses as part of the health extension programme. Under the iCCM programme, the health extension workers examine, classify and treat pneumonia.[34 35]

The OHEP intervention had three components, that is, community engagement activities, capacity building of health extension workers and women's development group leaders, and strengthening of the district health services' ownership and accountability of the primary newborn and child health services. The intervention was conducted in 26 intervention districts with 26 comparison districts spread in four regions of Ethiopia, namely Tigray, Amhara, Oromia and Southern Nations, Nationalities and Peoples Regions. The intervention started in 2016 and was completed in 2018. For the evaluation, baseline and end line surveys were performed. This paper reports a secondary analysis of end line cross-sectional data.[33]

### Participants
This study included all caregivers and children aged 2–59 months, who resided in 52 study districts. It also includes all health posts with one or two health extension workers per health post serving these families.

### Sampling
This study was based on secondary analysis of data from the endline survey that was part of the evaluation of the OHEP intervention. Sample size for the end line survey was estimated to measure changes in care-seeking and appropriate treatment for childhood illnesses between intervention and comparison areas at baseline and endline. Assumptions considered for the sample size calculation for the main OHEP evaluation[36] were 80% power to detect differences of 15% for the reported level of care-seeking (55%) and 20% for appropriate treatment for childhood illnesses (47%) at the baseline, design effect of 1.001% and 90% completeness. Accordingly, a sample size of 12 000 households was obtained. With this number of households, 6532 children below the age of 5 years were expected to be reached by the survey, of whom 368 were assumed to have any illnesses and 308 to have suspected pneumonia within 2 weeks before the survey.

The parent study used a sampling frame generated based on the 2007 Ethiopian Central Statistical Agency housing and population survey. Two hundred enumeration areas, that is, clusters, were selected from 52 study districts with probability proportional to size. A two-staged cluster sampling followed by systematic sampling to select 60 representative households from each study cluster. All caregivers of children aged 2–59 months living

in the selected households were interviewed. Moreover, all health posts and one or two health extension workers from each health post serving the population of the study clusters were included.[37]

## Data collection

Data were collected using structured and pre-tested interviewer-administered questionnaires through an electronic data collection software (CSpro survey software). The questionnaires were translated into local languages (Amharic, Tigrigna and Oromiffa) and back-translated into English. They comprised three main modules; household, healthcare provider and health facility modules (see online supplemental files 1–3). Data collectors and supervisors, who had bachelor's degree or above, were trained for 2 weeks on data collection techniques, procedures, quality assurance and ethical considerations of the study. Further detailed information about data collection and quality control is available in the published study protocol.[33]

## Outcomes

The outcomes of this study are caregivers' awareness of pneumonia treatment, care-seeking behaviour and coverage of pentavalent-3 immunisation as defined in the Demographic and Health Surveys.[28] The awareness of availability of pneumonia treatment was calculated as the proportion of caregivers who had heard messages regarding pneumonia treatment. Suspected pneumonia was ascertained by asking the caregiver if the child had cough combined with either fast or difficult breathing due to chest problems within 2 weeks before the survey. Care-seeking was defined as children with suspected pneumonia for whom advice or treatment was sought from an appropriate care provider, that is, either government or private providers. The vaccination status of children aged 12–23 months was primarily assessed by reviewing immunisation cards. When cards were not available at home, the caregivers were requested to report the type of vaccines their children had received. Hence, coverage of pentavalent vaccination was estimated as the proportion of children 12–23 months who had received three doses of pentavalent vaccine.

## Health postreadiness

The readiness of health posts for sick child care was assessed using the WHO Service Availability and Readiness Assessment tool.[38] Using 23 tracer items, the preparedness of facilities was shown in five domains or indices, that is, basic amenities, basic equipment, standard precaution equipment for infection prevention, diagnostic capacity and essential medicines. The mean availability of items across the four domains of readiness was estimated by assigning equal weight to each of the items, and was expressed as a proportion. Health posts' diagnostic capacity was shown as the proportion of facilities having rapid diagnostic test for malaria. The general service readiness was calculated as the average of percentages depicting mean availability

of tracer items in five indices.[38] The availability of vaccination card at the health posts was also estimated. The number of health extension workers working at the health post and the percentage of these workers trained in iCCM and who had received supportive supervision during 6 months before the survey were also calculated. The health post demand generation activities were recorded as the proportion of health posts showing opening days or that used community forums to deliver maternal and child health education.

## Analyses

The household socioeconomic status was constructed through principal component analysis of household assets, ownership of house, livestock, agricultural land and access to utilities and infrastructures. The factor scores were summed and ranked into quintiles from the poorest to the least poor.

The study linked the household, health facility and care provider information. Our analysis was based on linked samples for outcome variables, that is, caregivers' awareness of pneumonia treatment (n=4934), care-seeking when the child was sick (n=613) and vaccination of 12–23 months old children with a third dose of pentavalent immunisation (n=860). Care-seeking was assessed for all childhood illness episodes, including symptoms of suspected pneumonia as reported by caregivers for the 2 weeks prior to the survey. The effect of clustering on three of the study outcomes was examined by estimating intracluster correlation coefficients (ICC). A multilevel binary logistic regression model was fitted to examine the association between health post readiness and household level awareness, care-seeking and utilisation of three doses of pentavalent vaccinations. We checked for potential household-level confounders. The fitness of the model was checked through Likelihood Ratio Test. Data were analysed using Stata V.14.

## Patient and public involvement

Patients or the public were not involved in the design or conduct or reporting or dissemination plans of this research.

## RESULTS
### Characteristics of caregivers and children

A total of 10 785 households from 181 study clusters, 165 health posts and 274 health extension workers were included in the survey. A total of 4729 eligible caregivers with 5787 children aged 2–59 months were included in the analyses. A majority of the caregivers had no schooling (59.4%) and were currently married (88.6%). About two-thirds (64.0%) of caregivers were able to access the nearest health facility within 30 min of walk from their home (table 1).

### Characteristics of health posts and health extension workers

The median number of households served by the health post was 1181. The majority (85.1%) of the health posts

**Table 1** Sociodemographic characteristics of caregivers and children aged 2–59 months in four regions of Ethiopia, December 2018 to February 2019

| Characteristics | Frequency | Percentage |
|---|---|---|
| Caregivers' characteristics (n=4729) | | |
| Age | | |
| <25 | 885 | 18.7 |
| 25–29 | 1281 | 27.1 |
| 30–34 | 1038 | 22 |
| 35–39 | 867 | 18.3 |
| ≥40 | 658 | 13.9 |
| Marital status | | |
| Currently married | 4067 | 88.6 |
| Unmarried | 521 | 11.4 |
| Education | | |
| No schooling | 2810 | 59.4 |
| Schooling | 1919 | 40.6 |
| No of children under-5 | | |
| 1 | 3487 | 73.7 |
| 2 | 1148 | 24.3 |
| 3+ | 94 | 2 |
| Socioeconomic quintiles | | |
| Q1 (poorest) | 1024 | 21.7 |
| Q2 | 982 | 20.8 |
| Q3 | 874 | 18.5 |
| Q3 | 895 | 18.9 |
| Q5 (least poor) | 954 | 20.1 |
| Walking distance from household to nearest health facility (n=3918) | | |
| ≤30 min | 2507 | 64 |
| >30 min | 1411 | 36 |
| Child (2–59 months) characteristics (n=5787) | | |
| Sex | | |
| Male | 2961 | 51.2 |
| Female | 2826 | 48.8 |
| Age | | |
| 2–11 months | 959 | 16.6 |
| 12–23 months | 992 | 17.1 |
| 24–35 months | 1114 | 19.3 |
| 36–59 months | 2722 | 47 |

**Table 2** Characteristics of health posts in four regions of Ethiopia, December 2018 to February 2019

| Characteristics | Frequency | Percentage |
|---|---|---|
| Health posts catchment area population (n=165) | | |
| No of households, median (IQR) | 1181 (841–1572) | |
| No of children under 5 years, median (IQR) | 780 (493–1065) | |
| Health extension workers' characteristics (274) | | |
| No of health extension workers per health post | | |
| One | 35 | 12.7 |
| Two | 142 | 51.8 |
| Three and above | 97 | 35.4 |
| Health postopening days per week | | |
| 1–4 days | 41 | 14.9 |
| 5–7 days | 233 | 85.1 |
| Posted health postopening days | 54 | 19.7 |
| Trained for iCCM of childhood illnesses | 216 | 78.8 |
| Received supportive supervision in the last 6 months | 216 | 78.8 |
| Participated in Performance Review and Clinical Mentorship meetings | 126 | 46.0 |
| Used community forums to deliver maternal and child health education | 205 | 74.8 |

iCCM, integrated Community Case Management.

### Health postpreparedness to deliver sick child care services

The general service readiness of health posts (n=165) to deliver sick child health services was estimated at 62.8%. Half of the health posts had rapid diagnostic test for malaria, and the mean availability of essential medicines was 66.9%. Relatively higher mean availability (80.1%) of tracer items was shown for basic equipment, while the lowest (48.7%) was for availability of standard precaution items. Very few health posts had all essential medicines and standard precaution equipment. Most of the health posts (84.2%) had vaccination cards (figure 1).

### Awareness of treatment, actual care-seeking and utilisation of preventive immunisation

During the 2 weeks before the survey 704 (12.3%) of the children had any illnesses. Of these, 44 children had symptoms and signs of suspected pneumonia. Only one in five (23.9%) caregivers were aware of the availability of pneumonia treatment services. Healthcare was sought for one-third (36.1%) of children with any illnesses and for 56.8% of children with suspected pneumonia. Almost

were functionally open for 5 days or more per week. More than 1/10 (12.7%) of the health posts were served by only one health extension worker. Most (78.8%) of the health extension workers were trained in the iCCM of childhood illnesses. More than three-fourth (78.8%) had received supportive supervision within 6 months preceding the survey. Three-quarters used community forums or meetings to deliver maternal and child health education within 3 months prior to the survey (table 2).

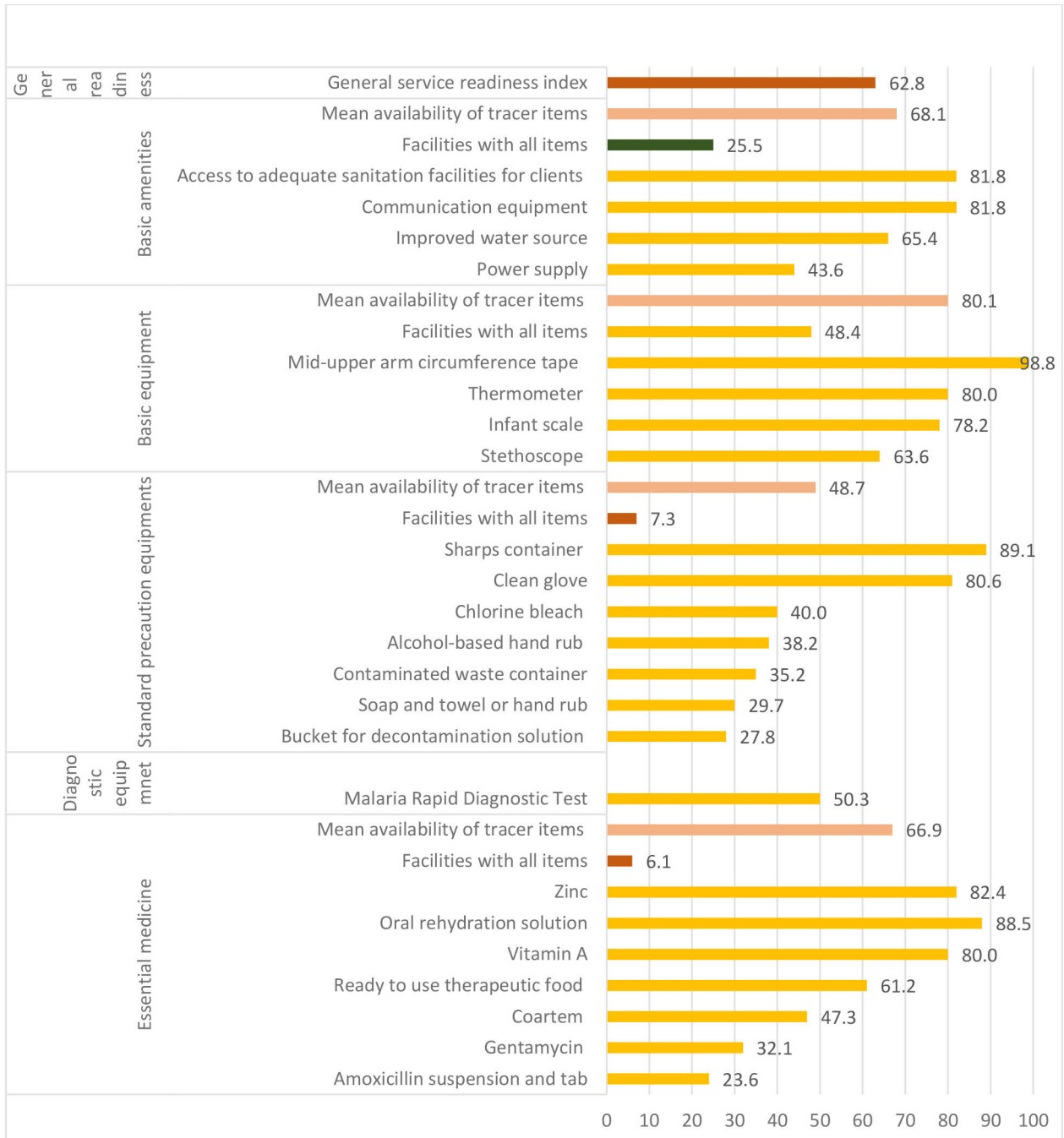

**Figure 1** Percentage of health posts (n=165) with drugs and supplies to deliver pneumonia-related and other sick child care services in four regions of Ethiopia, December 2018 to February 2019.

half (49.3%) of children aged 12–23 months had received three doses of pentavalent vaccinations (table 3).

### Association between health postreadiness and awareness and utilisation of pneumonia care

The ICC indicated that the study outcomes, that is, awareness of pneumonia treatment (ICC 0.29, 95% CI 0.24 to 0.36), care-seeking behaviour (ICC 0.16, 95% CI 0.10 to 0.27) and pentavalent vaccination (ICC 0.28, 95% CI 0.20, 0.38) significantly varied with level of clusters. While checking for the household-level confounders, we found that none of the household, caregiver and child

characteristics were associated with the outcomes and main exposure variables, that is, the five health postreadiness indices. But regardless of the statistical screening criteria (p<0.2), analyses were adjusted for maternal age, education and area of residence (intervention-comparison area) variables.

Analyses tested the association of general health post readiness index with study outcomes were adjusted for health extension workers' iCCM training, supportive supervision, participation at performance review and clinical mentorship meeting, home vising, use of

**Table 3** Childhood illness and care utilisation by child demographic characteristics in four regions of Ethiopia, December 2018 to February 2019

| Service utilisation | Frequency | Percentage |
|---|---|---|
| Children with any illnesses | | |
| Yes | 704 | 12.3 |
| No | 5021 | 87.7 |
| Childhood illnesses by sex (n=704) | | |
| Male | 362 | 51.4 |
| Female | 342 | 48.6 |
| Childhood illnesses by age (n=704) | | |
| 2–11 months | 121 | 17.2 |
| 12–23 months | 147 | 20.9 |
| 24–35 months | 152 | 21.6 |
| 36–59 months | 284 | 40.3 |
| Child with suspected pneumonia | | |
| Yes | 44 | 0.8 |
| No | 5743 | 99.2 |
| Child with suspected pneumonia treated with antibiotics (n=44) | | |
| Yes | 17 | 38.6 |
| No | 27 | 61.4 |
| Full pentavalent vaccination by sex (n=489) | | |
| Male | 258 | 52.8 |
| Female | 231 | 47.2 |

**Table 4** General health postreadiness and caregivers' awareness, care-seeking and utilisation of pentavalent-3 immunisation in four regions of Ethiopia, December 2018 to February 2019

| Awareness and utilisation | General readiness | |
|---|---|---|
| | Crude OR (95% CI) | Adjusted OR (95% CI) |
| Awareness of treatment service (N=4934)* | 0.9 (0.7 to 1.1) | 0.9 (0.7 to 1.1) |
| Care-seeking (N=613)* | 0.7 (0.5 to 0.9) | 0.6 (0.4 to 0.8) |
| Pentavalent-3 (N=860)* | 1.2 (0.9 to 1.6) | 1.2 (0.8 to 1.6) |

*Analyses adjusted for health extension workers' iCCM training, supportive supervision, participation at performance review and clinical mentorship meeting, home vising, use of community forums, opening days and number of staff at health post, mothers' age, education and area of residence.
iCCM, integrated Community Case Management .

suspected pneumonia. There was no consistent association between facility preparedness and utilisation of preventive and curative services.

## Strengths and limitations

With an attempt of narrowing the existing scarcity of evidence, our study examined the association between health post readiness and household-level awareness and utilisation of pneumonia relevant health services in a wider geographic area. Readiness of health posts was measured using the standard WHO Health Service Availability and Readiness Assessment tool. This tool is presumed to generate objective and reliable information that is comparable across or within countries. We pretested and adopted the tool to the local context and to level of care provided at health posts.[38] When vaccination cards were not available at home, children's pentavalent immunisation status was assessed through interviewing the caregivers. Ascertainment of childhood suspected pneumonia and other illnesses was based on the caregivers' 2 weeks reported symptoms prior to the survey. These ascertainment techniques have been used by the Demographic and Health Surveys, but might anyhow be influenced by recall bias.[28] Furthermore, we assessed health post service readiness and pneumonia service utilisation in selected districts of four Ethiopian regions. The findings may not be generalisable to other geographical areas and health system levels.

## Health postpreparedness

This study showed that two-thirds of the health posts were ready to provide sick child care, suggesting that the remaining facilities were not able to deliver such services. More or less similar level of structural preparedness of health posts or primary healthcare facilities for pneumonia and other sick child care were previously reported in Ethiopia and from other low-income and middle-income countries.[10 39 40] Furthermore, health posts or health centres of Ethiopia and other low-income

community forums, opening days and number of staff at health post, mothers' age, education and area of residence (intervention-comparison area) variables. The adjusted multilevel analyses revealed that general health post readiness was not associated with caregivers' awareness of availability of pneumonia treatment (adjusted OR, AOR 0.9, 95% CI 0.7 to 1.1) or utilisation of pentavalent-3 immunisation (AOR 1.2, 95% CI 0.8 to 1.6). The general health postreadiness was negatively associated with care-seeking for childhood illnesses (AOR 0.6, 95% CI 0.4 to 0.8)] (table 4).

As illustrated in table 5, none of the health post readiness indices were associated with caregivers' awareness of availability of pneumonia treatment and care-seeking for childhood illnesses. Only availability of standard precaution equipment for infection prevention was positively associated with utilisation of pentavalent-3 immunisation (AOR 4.5, 95% CI 1.6 to 12.8). Home visiting by the health extension workers was associated with higher odds for caregivers' awareness of availability of pneumonia treatment (AOR 2.9, 95% CI 2.3 to 3.6).

## DISCUSSION

Overall, this study showed insufficient health post service preparedness and low household awareness and utilisation of preventive and curative services for childhood

**Table 5** Health postpreparedness indices associated with caregivers' awareness, care-seeking and utilisation of three doses of pentavalent vaccines in four regions of Ethiopia, December 2018 to February 2019

| Characteristics | Awareness of treatment service (N=4934) | | | | Care-seeking (N=613) | | | | Pentavalent-3 (N=860) | | | |
|---|---|---|---|---|---|---|---|---|---|---|---|---|
| | Aware | Unaware | Crude OR (95% CI) | Adjusted OR (95% CI) | Sought care | Didn't seek care | Crude OR (95% CI) | Adjusted OR (95% CI) | Vaccinated | Not vaccinated | Crude OR (95% CI) | Adjusted OR (95% CI) |
| Basic amenities | | | | | | | | | | | | |
| All not available | 923 | 2867 | 1.0 | 1.0 | 161 | 270 | 1.0 | 1.0 | 325 | 337 | 1.0 | 1.0 |
| All available | 256 | 888 | 0.9 (0.6 to 1.5) | 0.9 (0.6 to 1.5) | 49 | 133 | 0.7 (0.4 to 1.1) | 0.7 (0.4 to 1.2) | 98 | 100 | 0.9 (0.6 to 1.8) | 0.9 (0.6 to 1.8) |
| Basic equipment | | | | | | | | | | | | |
| All not available | 590 | 2021 | 1.0 | 1.0 | 117 | 202 | 1.0 | 1.0 | 219 | 224 | 1.0 | 1.0 |
| All available | 589 | 1734 | 1.3 (0.9 to 1.9) | 1.2 (0.8 to 1.9) | 93 | 201 | 0.8 (0.5 to 1.3) | 0.9 (0.5 to 1.4) | 204 | 213 | 0.9 (0.6 to 1.6) | 0.8 (0.5 to 1.4) |
| Standard precaution equipment for infection prevention | | | | | | | | | | | | |
| All not available | 1091 | 3555 | 1.0 | 1.0 | 202 | 379 | 1.0 | 1.0 | 386 | 414 | 1.0 | 1.0 |
| All available | 88 | 200 | 1.9 (0.9 to 3.9) | 1.9 (0.9 to 4.2) | 8 | 24 | 0.6 (0.2 to 1.7) | 0.8 (0.3 to 2.1) | 37 | 23 | 2.3 (0.9 to 5.6) | 4.5 (1.6 to 12.8) |
| Rapid diagnostic test for malaria | | | | | | | | | | | | |
| No available | 597 | 1858 | 1.0 | 1.0 | NA | NA | NA | NA | NA | NA | NA | NA |
| Available | 582 | 1897 | 0.8 (0.5 to 1.2) | 0.7 (0.4 to 1.1) | NA | NA | NA | NA | NA | NA | NA | NA |
| Essential medicine | | | | | | | | | | | | |
| First tercile | 354 | 1025 | 1.0 | 1.0 | 58 | 98 | 1 | 1 | 108 | 126 | 1.0 | 1.0 |
| Second tercile | 396 | 1355 | 0.9 (0.5 to 1.4) | 0.9 (0.5 to 1.4) | 76 | 142 | 0.8 (0.4 to 1.5) | 0.8 (0.4 to 1.5) | 129 | 145 | 0.9 (0.5 to 1.8) | 0.8 (0.4 to 1.6) |
| Third tercile | 429 | 1375 | 0.8 (0.5 to 1.3) | 0.8 (0.5 to 1.4) | 76 | 163 | 0.7 (0.4 to 1.3) | 0.7 (0.4 to 1.4) | 186 | 166 | 1.3 (0.7 to 2.3) | 1.1 (0.6 to 2.1) |
| No of health extension workers per health post | | | | | | | | | | | | |
| One | 226 | 784 | 1.0 | 1.0 | 47 | 71 | 1.0 | 1.0 | 73 | 68 | 1.0 | 1.0 |
| Two and above | 953 | 2971 | 1.2 (0.7 to 1.9) | 1.2 (0.7 to 1.9) | 163 | 332 | 0.8 (0.4 to 1.4) | 0.8 (0.5 to 1.5) | 350 | 369 | 0.9 (0.5 to 1.6) | 0.7 (0.4 to 1.4) |
| No of health postopening days | | | | | | | | | | | | |
| Less than 5 days | 209 | 582 | 1.0 | 1.0 | 43 | 71 | 1.0 | 1.0 | 61 | 67 | 1.0 | 1.0 |
| Five days and above | 970 | 3173 | 0.8 (0.5 to 1.3) | 0.7 (0.4 to 1.2) | 167 | 332 | 0.9 (0.5 to 1.5) | 0.7 (0.4 to 1.4) | 362 | 370 | 1.3 (0.7 to 2.5) | 1.4 (0.7 to 2.8) |
| Health extension workers used community forum | | | | | | | | | | | | |
| No | 297 | 1003 | 1.0 | 1.0 | 70 | 139 | 1.0 | 1.0 | 113 | 131 | 1.0 | 1.0 |
| Yes | 882 | 2752 | 1.2 (0.8 to 1.8) | 1.3 (0.8 to 1.9) | 140 | 264 | 1.0 (0.6 to 1.6) | 0.9 (0.6 to 1.5) | 310 | 306 | 1.3 (0.8 to 2.2) | 1.3 (0.7 to 2.2) |
| Health extension workers received supervision | | | | | | | | | | | | |
| No | NA | NA | NA | NA | 44 | 105 | 1.0 | 1.0 | 60 | 68 | 1.0 | 1.0 |
| Yes | NA | NA | NA | NA | 166 | 298 | 1.3 (0.8 to 2.3) | 1.5 (0.8 to 2.6) | 363 | 369 | 1.4 (0.7 to 2.5) | 1.1 (0.5 to 2.2) |
| Health extension workers received iCCM training | | | | | | | | | | | | |
| Did not received training | NA | NA | NA | NA | 40 | 74 | 1.0 | 1.0 | NA | NA | NA | NA |

Continued

**Table 5** Continued

| Characteristics | Awareness of treatment service (N=4934) | | | | Care-seeking (N=613) | | | | Pentavalent-3 (N=860) | | | |
|---|---|---|---|---|---|---|---|---|---|---|---|---|
| | Aware | Unaware | Crude OR (95% CI) | Adjusted OR (95% CI) | Sought care | Didn't seek care | Crude OR (95% CI) | Adjusted OR (95% CI) | Vaccinated | Not vaccinated | Crude OR (95% CI) | Adjusted OR (95% CI) |
| Received training | NA | NA | NA | NA | 170 | 329 | 0.9 (0.5 to 1.6) | 0.9 (0.5 to 1.7) | NA | NA | NA | NA |
| Health extension workers participated at Performance Review and Clinical Mentorship meeting | | | | | | | | | | | | |
| No | NA | NA | NA | NA | 123 | 252 | 1.0 | 1.0 | 232 | 252 | 1.0 | 1.0 |
| Yes | NA | NA | NA | NA | 87 | 151 | 1.2 (0.8 to 1.9) | 1.2 (0.7 to 2.0) | 191 | 185 | 1.2 (0.7 to 1.9) | 0.9 (0.5 to 1.5) |
| Home vising | | | | | | | | | | | | |
| No | 934 | 3445 | 1 | 1 | 175 | 357 | 1 | 1 | 372 | 386 | 1.0 | 1.0 |
| Yes | 244 | 309 | 2.9 (2.4 to 3.7) | 2.9 (2.3 to 3.6) | 35 | 46 | 1.6 (0.9 to 2.6) | 1.5 (0.9 to 2.5) | 51 | 38 | 1.6 (0.9 to 2.7) | 1.6 (0.9 to 2.7) |

iCCM, integrated Community Case Management; NA, not available.

countries were found with low readiness to provide quality care to sick children.[9] According to the Ethiopia health system, a health centre is structured to support and strengthen five health posts within their catchment areas, hence insufficient preparedness of the surveyed health posts could be explained by scarcity of supplies at health centres.[29 41] The weak linkage and inadequate support from the health centres or the health system could further cause scarcity of drugs and supplies and unpreparedness of staff to serve at health posts.[22 42] The lack of readiness at health posts could also be related to the donor-dependent nature of supplies and the health extension workers' lack of accountability and capacity in supply-chain management.[34 43 44]

Inadequate readiness of health posts in the study setting and other low-income countries indicates a serious challenge to community case management of pneumonia, particularly in the rural or unreached communities, where a majority of preventable deaths occur.[6 28] Most importantly, only a few of the surveyed health posts had all essential medicines and just half had diagnostic equipment, clearly indicating their limitations in providing effective pneumonia or sick child treatment.[5 6] Scarcity of essential medicines in Ethiopia and other sub-Saharan African countries results in missed pneumonia treatments at facility level.[13 17 29] Unavailability of rapid diagnostic tests impairs community health workers' ability to differentiate suspected pneumonia from malaria in case of symptom overlap, a common clinical problem in African children.[45–47]

### Awareness and utilisation of pneumonia-related health services

Our study revealed low healthcare utilisation for pneumonia-specific preventive and curative services. These levels of service utilisation were lower compared with the reported regional pentavalent-3 immunisation coverage (80%) and care-seeking behaviour (85%) for childhood suspected pneumonia and other illnesses in sub-Saharan African Countries.[48 49] Community awareness of illness and sick child care is a prerequisite for timely utilisation of health services.[50] We found that less than a fifth of caregivers were aware of the availability of pneumonia treatment services, and this might partly explain the observed poor utilisation of pneumonia-related health services in the study setting.[51 52] The reported low care utilisation could also be explained by inadequate readiness or service quality of the primary healthcare facilities for pneumonia-related preventive and treatment services.[39 53] The OHEP evaluation studies have revealed a low quality of sick child care services provided at the primary healthcare facilities, and caregivers of children have also mentioned this as a key barrier to seek care at health posts.[54–56] A study in 22 African countries noted a low level of community trust in public health facilities as an important reason to the low coverage of child vaccination.[57]

## Association between health post preparedness and utilisation of preventive immunisation and care-seeking behaviour

Earlier studies have shown a positive association between health facility readiness and utilisation of first-level sick child care.[11 21] We did not find any consistent pattern of relationship between facility readiness indices and utilisation of services. There was a positive association between the availability of standard precaution equipment for infection prevention and utilisation of pentavalent-3 immunisation. However, the general health post readiness had no association with awareness and coverage of pentavalent-3 immunisation, but a negative association with care-seeking for childhood suspected pneumonia and other illnesses. Studies in Haiti and Ethiopia have shown absence of association between readiness of primary healthcare facilities for sick child care and caregivers' utilisation and satisfaction to the respective services.[40 58] The lack of consistent positive association may be linked to the dominating low level of health post preparedness. The community's value to quality of healthcare service is a key driver of their decision to seek care, and this may subserve the lack of consistent association between facility readiness and utilisation of pneumonia services. Irrespective of the readiness of health posts, caregivers' preconceived lack of trust to quality of primary child health services could motivate the use of other facilities with perceived higher service quality.[54] Equipping healthcare facilities with relevant equipment is a prerequisite to enhance the quality of iCCM services.[6] Hence, the reported lack of association of health post readiness with awareness and utilisation of pneumonia treatment services suggests that simply equipping facilities with necessary supplies is not a guarantee to reach the intended level of community awareness and utilisation of health services. Health facility strengthening efforts should go along with implementation of awareness creation and demand generation interventions to increase the community awareness, trust and utilisation of pneumonia related and other child health services.[50] Our previous study showed higher parents' care-seeking for childhood suspected pneumonia among those with improved awareness of treatment service.[59] The current study also illustrated that parents' awareness of pneumonia treatment was higher when health extension workers had visited at home. Home visiting by the community health workers is a vital strategy to promote child health and enhance awareness and utilisation of health services.[50 60] A substudy of the same project also showed that awareness creation and delivery of preventive child health interventions (such as immunisations) were the main components of outreach services delivered by the health extension workers.[54]

## Relation of findings with already published OHEP evaluation studies

This study was part of the end line evaluation of the OHEP intervention. The findings showed sizeable gaps in structural readiness of health posts for sick child care. Results of our published baseline study also illustrated the scarcity of essential drugs and other supplies at the health posts.[29] Substudies of the same project investigated quality of sick child care, showing low clinical performance of the health extension workers to identify and treat childhood suspected pneumonia and other illnesses.[55 56] A qualitative study noted lack of caregivers' trust in the health extension workers' clinical competency to manage sick children and a low availability of essential drugs, diagnostics and other supplies at health posts. The low quality of sick child care at health posts is a barrier to use these services.[54]

## CONCLUSIONS

This study has shown a low health post readiness for services, and low household awareness and utilisation of pneumonia-relevant preventive and curative services. Parents' awareness and utilisation of pneumonia-specific preventive and curative services were not consistently associated with the health post readiness. The results underline the critical importance of intensifying the health extension workers' awareness creation and demand generation efforts in each kebele (the lowest administrative unit in Ethiopia). Enhancing the coverage of home visiting and other awareness creation activities are crucial to boost community awareness and utilisation of pneumonia and other sick child care services. Our findings also underline the pivotal role of improving the availability and quality of pneumonia and other sick child care services to ensure optimal uptake of the services. It is imperative that the district health offices strengthen the linkages within the primary healthcare units to increase the availability of essential medicines and readiness of the health posts for sick child care. The office should also optimise the availability of essential medicines and supplies at health centres that are the suppliers to the satellite health posts. Improving the coverage of regular supportive supervision, performance reviews and clinical mentorship could also help to timely identify and solve gaps in the availability of drug and other supplies at health posts. Community awareness creation and demand generation efforts should simultaneously be accompanied with health facility strengthening strategies.

**Author affiliations**
[1]Department of Human Nutrition, Institute of Public Health, College of Medicine and Health Sciences, University of Gondar, Gondar, Amhara Region, Ethiopia
[2]Department of Epidemiology and Biostatistics and Department of Reproductive Health and Population, Addis Continental Institute of Public Health, Addis Ababa, Ethiopia
[3]School of Public Health, Addis Ababa University, Addis Ababa, Ethiopia
[4]Department of Health System and Policy, Institute of Public Health, College of Medicine and Health Sciences, University of Gondar, Gondar, Ethiopia
[5]London School of Hygiene and Tropical Medicine, London, UK
[6]Ethiopian Public Health Institute, Addis Ababa, Ethiopia

**Acknowledgements** We would like to forward our deepest gratitude to the study participants. Our special thanks to field assistants involved in the data collection process.

**Contributors** AT, LP, YB, YBO, AW and GAB contributed to the conceptualisation of the study. AT analysed and interpreted the data and drafted the manuscript. LP, YB,

YBO, AW and GAB contributed to analysis and writing of the paper. All authors have read and approved the final manuscript. YBO is the guarantor of the work.

**Funding** The study was funded by a grant from the Bill & Melinda Gates Foundation (grant INV-009691) to the London School of Hygiene & Tropical Medicine.

**Disclaimer** The funder had no role in data collection, analysis or interpretation of results.

**Competing interests** None declared.

**Patient and public involvement** Patients and/or the public were not involved in the design, or conduct, or reporting, or dissemination plans of this research.

**Patient consent for publication** Not applicable.

**Ethics approval** The original study was approved by the Ethical Review Boards of the Ethiopian Public Health Institute (protocol number SERO-012-8-2016), the London School of Hygiene & Tropical Medicine (protocol number 11235), and the University of Gondar (V/P/RCS/05/559/2019). A written informed consent was obtained from each household respondent, caregiver of the index child and the health workers.

**Provenance and peer review** Not commissioned; externally peer reviewed.

**Data availability statement** Data are available on reasonable request. The data for this manuscript were primarily collected by the Ethiopian Public Health Institute and London School of Hygiene & Tropical Medicine. Interested researchers may contact the focal person, YBO through email: Yemisrach.Okwaraji@lshtm.ac.uk. All requests will be reviewed by this committee and if granted, data will be shared without any identifiers.

**ORCID iDs**
Amare Tariku http://orcid.org/0000-0002-4939-3701
Lars Åke Persson http://orcid.org/0000-0003-0710-7954

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
