## [Reviewer comments · BMJ Open]

ARTICLE DETAILS

TITLE (PROVISIONAL)	Health post service readiness and use of preventive and curative services for suspected childhood pneumonia in Ethiopia: a cross-sectional study
AUTHORS	Tariku, Amare; Berhane, Yemane; Worku, Alemayehu; Biks, Gashaw; Persson, Lars; Okwaraji, Yemisrach

VERSION 1 – REVIEW

REVIEWER	Abubakari, Sulemana Kintampo Health Research Centre, Ghana Health Service
REVIEW RETURNED	22-Dec-2021

GENERAL COMMENTS	This study highlights health system challenges in LMICs and lack of community awareness of services available at health facilities. It is of great public health importance, and I recommend for publication although some minor revisions some of which are outlined below need to be considered. 1. Abstract, Design and setting: This part of the manuscript mainly focused on the design and does not include anything on the setting so the authors should revise this section to include the setting.2. Methods, Sampling: Authors should outline the assumptions being referred to in this section. This section should also be written to make it clearer. For example, it is not very clear what 'respectively' is referring to in this section.3. Methods, Analyses, Line 25, 'health' is wrongly spelt as 'heath'. Authors need to double check their spellings.
--

REVIEWER	Takahashi, K Teikyo University Graduate School of Public Health
REVIEW RETURNED	27-Jan-2022

GENERAL COMMENTS	The authors found the important finding that, in this study, there existed no association between facility readiness and awareness or utilisation of child health services. However, the contents itself is not sufficiently written. Especially, based on the findings, the authors should detail how to vitalize facility readiness and awareness or utilisation of child health services in the Ethiopian context. Especially, this reviewer is keen to know how the authors integrate the existing system with pneumonia and sick child care services at the health posts. Please detail on this matter citing references and existing bottleneck. If the authors address the comment above, this article will be much more interesting than current one with more and more international readers. Below are the comments for revision. 1.P8. ICCM Once you abbreviate the term, you don't have to write full spelling aside Abbreviation. See, "integrated community Case
---

	Management(iCCM)". 2.P10The Intra Cluster Correlation Coefficient should be shown with confidence intervals. 3.P14. "Strengths and limitations" should usually be put at the end of the discussion. In addition, the first three sentences read as the summary of findings, not strengths.The author should rewrite this section.
--	--

VERSION 1 – AUTHOR RESPONSE

Reviewer: 1

Dr. Sulemana Abubakari, Kintampo Health Research Centre

Comments to the Author:

This study highlights health system challenges in LMICs and lack of community awareness of services available at health facilities. It is of great public health importance, and I recommend for publication although some minor revisions some of which are outlined below need to be considered.

Response; Thank you!

1. Abstract, Design and setting: This part of the manuscript mainly focused on the design and does not include anything on the setting so the authors should revise this section to include the setting.

Response; Thank you. We have added '52 districts of four regions of Ethiopia', **Page#2, line 36-37.**

2. Methods, Sampling: Authors should outline the assumptions being referred to in this section. This section should also be written to make it clearer. For example, it is not very clear what 'respectively' is referring to in this section.

Response; Thank you. We have revised the text in these sections and corrected the misplaced "respectively". **Page#5.**

3. Methods, Analyses, Line 25, 'health' is wrongly spelt as 'heath'. Authors need to double check their spellings.

Response; Thanks. Corrected. **Page#7, line 195.** The entire body of text is edited for spelling errors.

Reviewer: 2

1. The authors found the important finding that, in this study, there existed no association between facility readiness and awareness or utilisation of child health services. However, the contents itself is not sufficiently written. Especially, based on the findings, the authors should detail how to vitalize facility readiness and awareness or utilisation of child health services in the Ethiopian context.

Response; Thank you for comments and suggestions. We have added further explanations to the low service preparedness at health post. We have supported these

arguments with new references #34, 43 & 44. Revisions are found in the Discussion **Page#14, line 299-302**.

Also, we have added explanations to the observed lack of association between health posts readiness and parents' awareness and utilization of pneumonia services, supported by references #40 and #54. **Page#16, line 337-341 & 347-349**.

Potential solutions to the low community awareness and utilization of pneumonia services and poor readiness of health posts were provided in the Conclusion section **Page#17, line 368-380**.

1. Especially, this reviewer is keen to know how the authors integrate the existing system with pneumonia and sick child care services at the health posts. Please detail on this matter citing references and existing bottleneck. If the authors address the comment above, this article will be much more interesting than current one with more and more international readers.

Response; Thank you for comments and suggestions. We have included additional information on the primary sick child care services as part of the health extension program. This information is available in the '**Study setting and design**' **Page#4**. We cited new references to support statements, i.e., ref.#34 & 35 (**Page#4, line 110-117**).

The bottlenecks or barriers for the implementation or utilization of iCCM services are already described in the introduction section with relevant citations **Page#4, line 86-93**.

1. Below are the comments for revision.
 - 1.P8. ICCM Once you abbreviate the term, you don't have to write full spelling aside Abbreviation. See, "integrated community Case Management(iCCM)".

Response; Thank you. Corrected.

- 3.1 P10 The Intra Cluster Correlation Coefficient should be shown with confidence intervals.

Response; Thank you. The 95% confidence intervals for the Intra Cluster Correlation Coefficients for three outcomes, i.e., parents' awareness of pneumonia treatment, care-seeking behaviour and pentavalent-3 immunization, are included in the result section, **Page#10, line 240-242**.

- 3.2 P14. "Strengths and limitations" should usually be put at the end of the discussion. In addition, the first three sentences read as the summary of findings, not strengths. The author should rewrite this section.

Response; Thank you. The authors guide of the BMJ Open requires bullets of "Strengths and limitations" next to the Abstract. The strengths and limitation of the study are also discussed in the discussion section. We notice that in the latest issues of BMJ Open, strengths and limitations are found either immediately after the initial summary in the discussion section or just before a concluding paragraph. We prefer the former, so that the reader is aware of strengths and weaknesses before entering into the discussion of findings. We prefer to keep the first sentence in this paragraph, since it points at a strength in relation to the current knowledge base. Some more information showing the strength of using the WHO instrument is also added in the discussion (**Page#14, line 280-282**).

VERSION 2 – REVIEW

REVIEWER	Takahashi, K
----------	--------------

	Teikyo University Graduate School of Public Health
REVIEW RETURNED	10-Mar-2022

GENERAL COMMENTS	The authors addressed properly to my comments. I believe that this article is ready for publication.
--